# Quasipseudometric Value Functions with Dense Rewards

## Abstract

As a generalization of reinforcement learning (RL) to parametrizable goals, goal conditioned RL (GCRL) has a broad range of applications, particularly in challenging tasks in robotics. Recent work has established that the optimal value function of GCRL $Q^*(s, a, g)$ has a quasipseudometric structure, leading to targetted neural architectures that respect such structure. However, the relevant analyses assume a sparse reward setting—a known aggravating factor to sample complexity. We show that the key property underpinning a quasipseudometric, viz., the triangle inequality, is preserved under a dense reward setting as well, specifically identifying the key condition necessary for triangle inequality. Contrary to earlier findings where dense rewards were shown to be detrimental to GCRL, we conjecture that dense reward functions that satisfy this condition can only improve, never worsen, sample complexity. We evaluate this proposal in 12 standard benchmark environments in GCRL featuring challenging continuous control tasks. Our empirical results confirm that training a quasipseudometric value function in our dense reward setting indeed either improves upon, or preserves, the sample complexity of training with sparse rewards. This opens up opportunities to train efficient neural architectures with dense rewards, compounding their benefits to sample complexity.

**Keywords:** Goal conditioned reinforcement learning, reward shaping, quasipseudometric value functions

## 1 Introduction

Reinforcement learning (RL) is a popular class of techniques for training autonomous agents to behave (near-)optimally, often without requiring a model of the task or environment. In goal-achieving tasks, traditional RL learns policies that reach a single goal at the minimum (maximum) expected cost (value) from any state. Contrastingly in multi-task settings, a goal conditioned value function models the cost-to-go to a *set of goal states*, not just one. This generalization from a single-goal case to goal-conditioned RL (GCRL) yields effective representations—powered by deep neural networks—for value functions capable of capturing abstract concepts underlying goal achievement in many complex tasks (M. Liu et al., 2022; Plappert et al., 2018; Wang et al., 2023).

Recent work has established that the true optimal value function in GCRL is always a *quasipseudometric*, i.e., a metric without the constraint of being symmetric, but crucially respecting the triangle inequality (B. Liu et al., 2023; Pitis et al., 2020; Wang & Isola, 2022). This allows the search for value functions to be naturally restricted to the space of quasipseudometrics. Additionally, such functions are designed to be *universal value function approximators* (UVFA), i.e., capable of approximating arbitrarily complex value functions. Accordingly, B. Liu et al. (2023) propose the metric residual network (MRN) architecture for GCRL value functions that explicitly accommodate an asymmetric component while maintaining the UVFA property and the triangle inequality. This and other similar approaches search a smaller subset of the space of value functions, yet the true optimal value function is guaranteed to reside in it. This has led to significant gains in terms of sample efficiency in recent GCRL advancements (B. Liu et al., 2023; Wang & Isola, 2022; Wang et al., 2023).

In this paper, we review some of the theoretical analyses underlying much of the work cited above. In particular, the proof of the key property of triangle inequality in B. Liu et al. (2023) is established for a

*sparse reward* setting that is easy to design but hard to learn from. By contrast, *dense reward* settings using various mechanisms, e.g., reward shaping, intrinsic motivation, human feedback etc., are generally known to improve sample efficiency. If dense reward-based value functions were to satisfy the triangle inequality, then their reward bias could be combined with the representational bias of quasipseudometrics to deliver a double punch to sample complexity. Furthermore in GCRL, since a goal is given, it may be easy to design a dense reward scheme parametrized by the goal that neither requires onerous, intricate reward engineering, nor is computationally demanding. This could bring the benefit of dense rewards–viz., reduced sample complexity–to GCRL with less human effort than in other RL tasks. However, existing negative results (Plappert et al., 2018) specifically in GCRL show that dense rewards significantly deteriorate the performance of state-of-the-art RL methods, and might appear to foreclose a discussion on their efficacy in GCRL. Contradictorily, we show that dense rewards can indeed bring their benefit to bear in GCRL as long as they satisfy a condition under which the triangle inequality is preserved for the *optimal* value function. Furthermore, we establish a condition under which the triangle inequality is preserved for *on-policy* value functions that may be encountered during RL iterations. This result adds nuance to recent contradictory finding (Wang et al., 2023) that on-policy value functions do *not* satisfy the triangle inequality. We show experiments in 12 benchmark GCRL tasks to establish that dense rewards indeed improve sample complexity in some tasks, but does not deteriorate sample efficiency in any task.

Our main contributions can be summarized as:

- We show that using rewards shaped with potential functions that serve as admissible heuristics, the optimal value function does satisfy the triangle inequality;

- We define and delineate a progressive criterion for GCRL policies and show that under such policies the on-policy value function satisfies the triangle inequality;

- Via experiments in 12 standard benchmark GCRL tasks, we show that dense rewards improve sample complexity as well as the learned policy in 4 of the 12 tasks, while not deteriorating performance in any task.

## 2 Background

This section covers the preliminaries on goal conditioned RL, the prevalent solution approaches for GCRL, and the recent architecture of metric residual networks that we use in this paper.

### 2.1 Goal-conditioned RL

Goal conditioned RL is modeled by goal-conditioned Markov decision process, $M = (\mathcal{S}, \mathcal{A}, \mathcal{G}, T, R, \gamma, \rho_0, \rho_{\mathcal{G}})$. While $\mathcal{S}, \mathcal{A}, T, \rho_0$ define the state action spaces, the transition function and the initial state distribution just like a standard MDP, $\mathcal{G}$ gives the space of goal states, and $\rho_{\mathcal{G}}$ is the distribution from which a goal is sampled at the beginning of an episode. Further, the reward function $R$ is additionally parametrized by the goal, $R : \mathcal{S} \times \mathcal{A} \times \mathcal{G} \mapsto \Re$. In the *sparse reward* setting, $R$ is often defined as

$$R(s, a, g) = \begin{cases} 0 & \text{if } M(s, a) = g \\ -1 & \text{otherwise} \end{cases} \tag{1}$$

where $M : \mathcal{S} \times \mathcal{A} \mapsto \mathcal{G}$ maps the product space of $\mathcal{S}$ and $\mathcal{A}$ to $\mathcal{G}$, essentially flagging goal achievement. As opposed to the common assumption $\mathcal{G} \subset \mathcal{S}$, $M$ allows action $a$ to decide whether the goal is reached (B. Liu et al., 2023). The agent's decision function is its *policy*, $\pi$, which can be either deterministic ($\pi : \mathcal{S} \times \mathcal{G} \mapsto \mathcal{A}$) or stochastic ($\pi : \mathcal{S} \times \mathcal{A} \times \mathcal{G} \mapsto [0, 1]$) yielding the probability of selecting actions.

### 2.2 Solution Approach: DDPG+HER

A popular approach to solving GCRL is a combination of off-policy actor-critic, e.g., DDPG (Lillicrap et al., 2016) with hindsight experience replay (HER) (Andrychowicz et al., 2017). DDPG in GCRL estimates a

goal conditioned critic for policy $\pi$

$$Q^\pi(s, a, g) = \mathbb{E}_{\pi, T, R} \left[ \sum_{t=0}^\infty \gamma^t r_{t,g} | s_0 = s, a_0 = a, g \right]$$

where the expectation is taken over future steps of rewards generated by the policy ($\pi$), and the $T, R$ functions. The critic is updated by minimizing the mean squared TD error over samples $(s_t, a_t, s_{t+1}, g)$ drawn from a replay buffer $D$,

$$L(Q) = \mathbb{E}_D \left[ (r_{t,g} + \gamma Q(s_{t+1}, \pi(s_{t+1}), g) - Q(s_t, a_t, g))^2 \right]. \tag{2}$$

By ensuring that $Q$ is differentiable w.r.t. actions $a$, the actor policy $\pi$ is updated in the direction of the gradient $\mathbb{E}[\nabla_{a_t} Q(s_t, a_t, g)]$, where the expectation is again evaluated using samples drawn from $D$. As these samples are drawn from state distributions generated by policies different from $\pi$, DDPG is an off-policy method, although it estimates Q-values in an on-policy way (Eq. 2). This last aspect will be scrutinized further in Sec. 3.2.

Hindsight experience replay (HER) (Andrychowicz et al., 2017) mitigates the sparse reward problem by relabeling failed trajectories. Instead of treating all experience traces where the agent failed to achieve a goal as is, HER changes the goal in some of them to match a step of the trace in hindsight—essentially pretending as if the agent's goal all along was to reach the state that it actually did. This transforms some of the failed episodes into successful experiences that are informative about goal achievement, and allows the agent to generalize, eventually, to the true goal distribution $\rho_\mathcal{G}$. A more recent method, goal-conditioned supervised learning (Ghosh et al., 2021), uses the idea of HER but in the context of offline RL. Both DDPG and HER remain popular and are commonly used in online GCRL (B. Liu et al., 2023).

## 2.3 Metric Residual Network

B. Liu et al. (2023) propose a novel neural architecture for GCRL critic based on the insight that the optimal negated action-value function, $-Q^*(s, a, g)$, satisfies the triangle inequality *in the sparse reward setting* of Eq. 1. Consequently, they introduce the metric residual network (MRN) that decomposes $-Q$ into the sum of a metric and an asymmetric residual component that provably approximates any quasipseudometric. Specifically,

$$Q(s, a, g) = -(d_{sym}(h_{sa}, h_{sg}) + d_{asym}(h_{sa}, h_{sg})) \tag{3}$$

where $h_{sa}$ and $h_{sg}$ are latent encodings of concatenated $(s, a)$ and $(s, g)$, $d_{sym}$ and $d_{asym}$ are symmetric and asymmetric distance components given by

$$d_{sym}(x, y) = \|\mu_1(x) - \mu_1(y)\|, \tag{4}$$
$$d_{asym}(x, y) = \max_i (\mu_{2i}(x) - \mu_{2i}(y))_+, \tag{5}$$

$\|.\|$ is the Euclidean distance, $(x)_+ = \max(x, 0)$ is also known as the rectified linear unit (ReLU) function, $\mu_1$ and $\mu_2$ are neural networks that map their inputs to vectors, and $\mu_{2i}$ is the $i$-th component of $\mu_2$'s output vector. Note from Eqs. 4, 5 that indeed $d_{sym}(x, y) = d_{sym}(y, x)$ but possibly $d_{asym}(x, y) \neq d_{asym}(y, x)$. The main idea is that Eq. 3 is a valid decomposition since $Q(s, a, g)$ is essentially a distance and any distance can be (in general) split into symmetric and asymmetric components, where either components can be 0 in limit cases. On the one hand, the Max-ReLU formulation of $d_{asym}$ in Eq. 5 is proven to ensure the UVFA property of Eq. 3 (B. Liu et al., 2023). On the other hand, $d_{sym}$ improves sample efficiency due to its symmetry. We use DDPG+HER with MRN critic architecture as the base GCRL method for this paper.

## 2.4 The Quasipseudometric Property

A space $\mathcal{X}$ with a distance measure $d$ defines a quasipseudometric, $(d, \mathcal{X})$, if (1) $\forall x \in \mathcal{X}, d(x, x) = 0$ and (2) $\forall x, y, z \in \mathcal{X}, d(x, y) + d(y, z) \geq d(x, z)$, also known as the *triangle inequality*. A metric is a special case of a

quasipeudometric that is also symmetric, but symmetry is not necessary in robotics problems, e.g., a robotic arm could rotate in one direction but not in the other leading to $d(x, y) \neq d(y, x)$. Additionally, a quasimetric is a special case of a quasipseudometric where it is necessary that $x \neq y \implies d(x, y) > 0$. This is also not necessarily true in robotics problems, for instance, when rotating a robotic arm, different quaternions $x \neq y$ could represent the same rotation, i.e., $d(x, y) = 0$. B. Liu et al. (2023) argue that in general GCRL problems, $-Q(s, a, g)$ may not be purely symmetric (i.e., it may have an asymmetric component), or $> 0$ for different states. As a result, the most general form of $-Q(s, a, g)$ for GCRL is neither a metric nor a quasimetric, but a quasipseudometric as constructed via Eqs. 3, 4, 5.

B. Liu et al. (2023) establish the critical property (2) of a quasipseudometric, i.e., the triangle inequality for optimal $-Q^*(s, a, g)$. This property is the key focus of this paper.

## 3 Triangle Inequality

In this section, we establish that both the optimal value function as well as on-policy value functions satisfy the triangle inequality under novel conditions.

### 3.1 Optimal Value Function

Our primary claim is that $-Q^*$ satisfies the triangle inequality not only in the sparse reward setting, but also in the presence of dense rewards, particularly potential shaped rewards (Ng et al., 1999). This observation lends GCRL to improved sample efficiency when approximating $-Q^*$ using a combination of MRN and potential shaped rewards.

We use the standard potential based shaping rewards (Ng et al., 1999) expressed as the difference between discounted potential ($\phi$) of the next state-action pair and the potential of the current state-action pair, for the same goal:

$$F(s, a, s', a', g) = \gamma\phi(s', a', g) - \phi(s, a, g) \tag{6}$$

and a simple potential function

$$\phi(s, a, g) = -\left(\frac{1 - \gamma^{d(s,a,g)/\eta}}{1 - \gamma}\right)$$

where $d$ is a distance measure between the state and the goal, and $\eta$ is a measure of the atomicity of actions—distance covered per time step. The specific form of $\phi$ shown above is explained later in the context of Obs. 1. The function $d$ must be such that

$$M(s, a) = g \implies d(s, a, g) = 0 \implies \phi(s, a, g) = 0 \tag{7}$$

i.e., the potential is maximized when the goal is reached. Therefore, Eq. 6 can be viewed as a heuristic measure of the agent's step-improvement in the desirability of its state with respect to its goal, where "desirability" is roughly the proximity of the goal. Thus $F$ is a suitable candidate to be an immediate step-reward, and is typically added to the default reward. An important property of such potential shaped reward function is that it leaves the optimal policy unchanged (Ng et al., 1999). Now, in the reward regime of Eq. 1,

$$Q^*(s, a, g) = -\left(\frac{1 - \gamma^{L^*(s,a,g)}}{1 - \gamma}\right)$$

where $L^*(s, a, g)$ is the *optimal* expected number of steps required to reach the goal $g$ from state $s$. Note the similarity of this expression of $Q^*$ to the potential function $\phi$ shown above. Our comparable choice of $\phi$ leads to a simple form of the condition that we show in the next section (Eq. 8) to be the key to satisfying the triangle inequality. Below, we formally specify its premise as an observation.

**Observation 1.** *If $d(s, a, g) \leq \eta L^*(s, a, g)$, then $\phi(s, a, g) \geq Q^*(s, a, g), \forall s, a, g$*

In other words, if $d(s, a, g)/\eta$ is an underestimate of $L^*$ then the above condition will be satisfied. Thus, $d$ acts as an admissible heuristic. In this paper, we study two simple forms of $d$: the arc-cosine distance

$d_{ac}(s, a, g) = \cos^{-1}\left(\frac{M(s,a) \cdot g}{\|M(s,a)\|\|g\|}\right)/\pi$, and the Euclidean distance $d_E(s, a, g) = \|M(s, a) - g\|$, both of which are metrics and satisfy Eq. 7 although $d_E$ is not necessarily bounded. These choices are simple and avoid intricate, environment-specific reward engineering, while producing underestimates of $L^*$ (for appropriate values of $\eta$). However, these choices are not necessary for our theoretical results to hold and any definition of $d$ that ensures $\phi \geq Q^*$ could be used. Many of the experimental tasks contain state representations that include angles, albeit mixed with non-angular features. Consequently, $d_{ac}$ may be relatively more accurate, yielding a dominant admissible heuristic.

We distinguish $Q^*(s, a, g)$—the optimal action values with unshaped sparse rewards—from $Q_F^*(s, a, g)$ which corresponds to action values with rewards shaped by $F$ in Eq. 6. Specifically,

$$Q^*(s, a, g) = \max_\pi \mathbb{E}_{\pi, T, R}\left[\sum_{t=0}^\infty \gamma^t r_{t,g} | s_0 = s, a_0 = a, g\right]$$

$$Q_F^*(s, a, g) = \max_\pi \mathbb{E}_{\pi, T, R}\left[\sum_{t=0}^\infty \gamma^t (r_{t,g} + F_{t,g}) | s_0 = s, a_0 = a, g\right].$$

Next we establish the validity of triangle inequality with $Q_F^*$ in two cases: (i) $\mathcal{G} \equiv \mathcal{S} \times \mathcal{A}$ and (ii) $\mathcal{G} \not\equiv \mathcal{S} \times \mathcal{A}$.

### 3.1.1 Case I: $\mathcal{G} \equiv \mathcal{S} \times \mathcal{A}$

In this setting, $M$ is the identity mapping. We use the notation $x_t = (s_t, a_t)$. The main result is:

**Proposition 1.** *Consider the shaped, goal-conditioned MDP $M_{GCF} = (\mathcal{S}, \mathcal{A}, \mathcal{G}, T, R + F, \gamma, \rho_0, \rho_g)$, with $\mathcal{G} \equiv \mathcal{S} \times \mathcal{A}$. The optimal universal value function $Q_F^*$ satisfies the triangle inequality: $\forall x^1, x^2, x^3 \in \mathcal{X}$,*

$$Q_F^*(x^1, x^2) + Q_F^*(x^2, x^3) \leq Q_F^*(x^1, x^3),$$

*The only condition $\phi$ must satisfy is*

$$\phi(s, a, g) \geq Q^*(s, a, g), \forall s, a, g \tag{8}$$

*w.r.t. the unshaped value function, for which a sufficient condition is established in Obs. 1.*

**Proof:** As in (B. Liu et al., 2023), consider the Markov policies $\pi_1, \pi_2, \pi_3$ that are optimal w.r.t. $Q_F^*(x^1, x^2)$, $Q_F^*(x^2, x^3)$, $Q_F^*(x^1, x^3)$ and the (non-Markov) policy $\pi_{1\rightarrow 2}$ defined for $t > 0$ as:

$$\pi_{1\rightarrow 2}(a|s_t) = \begin{cases} \pi_1(a|s_t), & x^2 \notin x_{<t} \text{ (set of states encountered prior to step t)} \\ \pi_2(a|s_t), & \text{otherwise.} \end{cases}$$

Let $\tau$ be the random variable that indicates the first time $\pi_{1\rightarrow 2}$ reaches $x^2$. In the steps below, we notate $F(s_t, a_t, s_{t+1}, a_{t+1}, g)$ as $F_{t,g}$, and $\mathbb{E}_{(x_t, r_t) \sim \pi, T, R, \tau}$ as $\mathbb{E}_{\pi,.}$ for brevity. Then define

$$q_{1\rightarrow 2}^1 = \mathbb{E}_{\pi_{1\rightarrow 2},.}\left[\sum_{t=0}^\tau \gamma^t (r_{t,g} + F_{t,g}) | x_0 = x^1, g = x^2\right],$$

$$q_{2\rightarrow 3}^2 = \mathbb{E}_{\pi_{1\rightarrow 2},.}\left[\sum_{t=\tau}^\infty \gamma^t (r_{t,g} + F_{t,g}) | x_\tau = x^2, g = x^3\right]$$

$$= \mathbb{E}_{\pi_{1\rightarrow 2},.}\left[\sum_{t=\tau}^\infty (\gamma^t r_{t,g}) + 0 - \gamma^\tau \phi_\tau\right].$$

Now,

$$Q_F^*(x^1, x^2) = \mathbb{E}_{\pi_1,.}\left[\sum_{t=0}^\tau \gamma^t (r_{t,g} + F_{t,g}) | x_0 = x^1, g = x^2\right] +$$

$$\mathbb{E}_{\pi_1,.}\left[\sum_{t=\tau+1}^\infty \gamma^t (r_{t,g} + F_{t,g}) | x_{\tau+1} = x^2, g = x^2\right], \tag{9}$$

and $\pi_1 \equiv \pi_{1 \to 2}$ for the first $\tau$ steps. Therefore, $Q_F^*(x^1, x^2) - q_{1 \to 2}^1$

$$= \mathbb{E}_{\pi_1, \cdot}\left[\sum_{t=\tau+1}^{\infty} \gamma^t(r_{t,g} + F_{t,g})|x_\tau = x^2, g = x^2\right],$$

$$= \mathbb{E}_{\pi_1, \cdot}\left[\sum_{t=\tau+1}^{\infty} (\gamma^t r_{t,g}) + \gamma^\infty \phi(.) - \gamma^{\tau+1}\phi_{\tau+1}|x^2, x^2\right]$$

$$= \mathbb{E}_{\pi_1, \cdot}\left[\sum_{t=\tau+1}^{\infty} (\gamma^t r_{t,g}) - \gamma^{\tau+1}\phi_{\tau+1}|x^2, x^2\right]$$

$$= \mathbb{E}_\tau[\gamma^{\tau+1}][Q^*(x^2, x^2) - \phi(x^2, x^2)]$$

$$\leq 0 \text{ by assumption (Eq. 8)}.$$

Similarly,

$$Q_F^*(x^2, x^3) = \mathbb{E}_{\pi_2, \cdot}\left[\sum_{t=0}^{\infty} \gamma^t(r_{t,g} + F_{t,g})|x_0 = x^2, g = x^3\right]$$

$$\leq \gamma^\tau \mathbb{E}_{\pi_2, \cdot}\left[\sum_{t=0}^{\infty} \gamma^t(r_{t,g} + F_{t,g})|x^2, x^3\right]$$

$$= \mathbb{E}_{\pi_2, \cdot}\left[\sum_{t=0}^{\infty} \gamma^{t+\tau}(r_{t,g} + F_{t,g})|x^2, x^3\right]$$

$$= \mathbb{E}_{\pi_2, \cdot}\left[\sum_{k=\tau}^{\infty} \gamma^k r_{k,g} + 0 - \gamma^\tau \phi_\tau|x^2, x^3\right]$$

$$= \mathbb{E}_{\pi_{1 \to 2}, \cdot}\left[\sum_{k=\tau}^{\infty} \gamma^k r_{k,g} - \gamma^\tau \phi_\tau|x^2, x^3\right]$$

$$= q_{2 \to 3}^2, \tag{10}$$

since $\pi_2 \equiv \pi_{1 \to 2}$ after $\tau$. Therefore, $Q_F^*(x^2, x^3) - q_{2 \to 3}^2 \leq 0$. Consequently,

$$Q_F^{\pi_{1 \to 2}}(x^1, x^3) = q_{1 \to 2}^1 + q_{2 \to 3}^2 \geq Q_F^*(x^1, x^2) + Q_F^*(x^2, x^3).$$

But since the optimal $Q_F^*(x^1, x^3) \geq Q_F^{\pi_{1 \to 2}}(x^1, x^3)$, we arrive at the triangle inequality. $\qquad \square$

### 3.1.2 Case II: $\mathcal{G} \not\equiv \mathcal{S} \times \mathcal{A}$

The proof of this case closely resembles (B. Liu et al., 2023); we highlight the main difference in blue color but also provide the rest of the proof for completeness. In this case, $M$ is an onto mapping. Given a goal $g$, the GCRL problem effectively reduces to a standard MDP and there exists a deterministic optimal policy $\pi^*$ for reaching the goal $g$ from an initial state $x = (s_0, a_0)$. Then, under deterministic dynamics,

$$Q_F^*(x, g) = \sup_{x': M(x')=g} Q_F^*(x, x').$$

Assuming the supremum is attainable, let

$$x_g = \arg\max_{x': M(x')=g} Q_F^*(x, x'), \tag{11}$$

then $Q_F^*(x, g) = Q_F^*(x, x_g)$. Assume for contradiction that this is not the case, i.e., $Q_F^*(x, g) \neq Q_F^*(x, x_g)$. There are two possibilities:

- If $Q_F^*(x, x_g) > Q_F^*(x, g)$: This would imply that by using a policy that selects $x_g$ rather than $g$, one could achieve a higher return. This contradicts the definition of $\pi^*$ as the optimal policy, thus $Q_F^*(x, x_g) > Q_F^*(x, g)$ cannot be true.

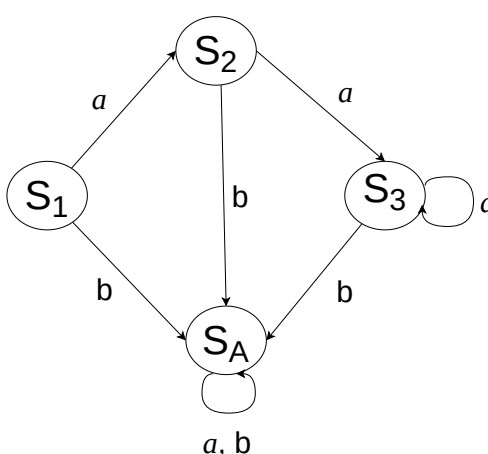

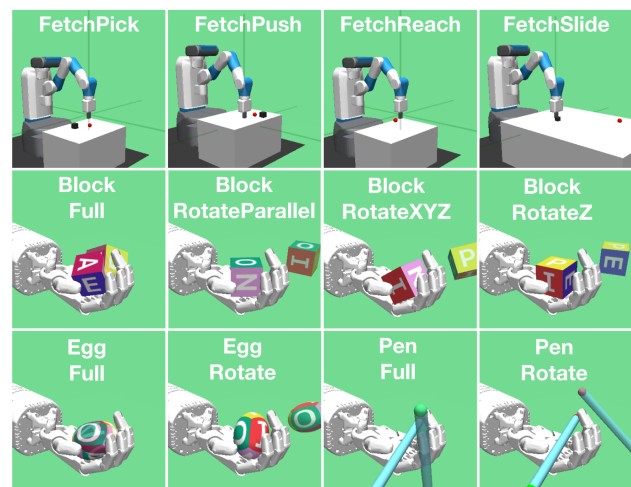

Figure 1: Counterexample to prove that policy invariance of potential based shaping does not trivially preserve triangle inequality.

Figure 2: GCRL benchmark environments (Plappert et al., 2018). Figure from (B. Liu et al., 2023).

- If $Q_F^*(x, x_g) < Q_F^*(x, g)$: Let

$$\tau = \min_t(M(x_t) = g),$$

such that $x_\tau$ is the first $(s, a)$ pair along the optimal trajectory that achieves the goal. There are two further cases:

1. After reaching $x_\tau$, $\pi^*$ will repeatedly return to $x_\tau$. In this case, we have $Q_F^*(x, x_g) \geq Q_F^*(x, x_\tau)$ by the definition of $x_g$ (Eq. 11) and

$$\begin{aligned} Q_F^*(x, x_\tau) &= Q_F^*(x, g) - \gamma^{\tau+1} Q_F^*(x_\tau, g) \\ &= Q_F^*(x, g) - \gamma^{\tau+1} \left[ Q^*(x_\tau, g) - \phi(x_\tau, g) \right] \\ &\geq Q_F^*(x, g), \text{ by Eq. 8.} \end{aligned} \tag{12}$$

Combining the two, we get $Q_F^*(x, x_g) \geq Q_F^*(x, g)$ which contradicts our assumption that $Q_F^*(x, g) > Q_F^*(x, x_g)$.

2. $\pi^*$ never returns to $x_\tau$ after reaching it for the first time. In this case, one can find the next $\tau' = \min_{t>\tau}(M(x_t) = g)$, such that $x_{\tau'}$ is another $(s, a)$ along the optimal trajectory. Again, there are two sub-cases:

   (a) If $\pi^*$ repeatedly visits $x_{\tau'}$, then the argument in the first case applies.

   (b) Otherwise, recursively find the next $\tau''$, and so on. Eventually, we may have a last state $x_\zeta$ such that no $t > \zeta$ satisfies $M(x_t) = g$. Then, $Q_F^*(x, x_g) \geq Q_F^*(x, x_\zeta) \geq Q_F^*(x, g)$. The last inequality is derived in the same way as Eq. 12. Alternatively, there may exist an infinite sequence of such $\{x_\tau\}$. Following this sequence, the claim remains true but the supremum is not attainable. However, in this case an $x_\tau$ can be found in the sequence such that $Q_F^*(x, x_\tau)$ is arbitrarily close to $Q_F^*(x, g)$. $\qquad \square$

### 3.1.3 Insufficiency of Policy Invariance

A well-known property of potential-based reward shaping is that it does not change the optimal policy (Ng et al., 1999). Could the preservation of the quasipseudometric property of the optimal value function under this strategy for reward densification be a trivial consequence of this property? We show in this section that that is not the case. Specifically, we show that violation of Eq. 8 can break triangle inequality without affecting policy invariance.

For a counterexample, consider the simple MDP shown in Fig. 1. It has 3 states, $S_1 - S_3$ navigable by action $a$ and an absorbing state $S_A$ that can only be reached by action $b$ from any other state.

**Proposition 2.** *In the counterexample of Fig. 1, the potential function for various states with goal $S_2$ or $S_3$ can be set such that policy invariance is preserved, yet the assumption of Eq. 8 as well as the quasipseudometric property are violated.*

**Proof:** First note that the optimal policy for goal $S_2$ or $S_3$ is to take action $a$, as these goals are unreachable via action $b$. For some $\alpha > 0$, we set:

$$\phi(S_1, a, S_3) = Q^*(S_1, a, S_3) + \alpha \qquad \text{(satisfies Eq. 8)}$$
$$\phi(S_2, a, S_3) = Q^*(S_2, a, S_3) - \alpha \qquad \text{(violates Eq. 8)}$$
$$\phi(S_1, a, S_2) = Q^*(S_1, a, S_2) + \alpha \qquad \text{(satisfies Eq. 8)}$$

These yield: $Q^*_F(S_1, a, S_3) = Q^*_F(S_1, a, S_2) = -\alpha$, $Q^*_F(S_2, a, S_3) = \alpha$. Since the goals are unreachable via action $b$, it is straightforward to set $\phi(*, b, *)$ such that the optimal policy stays unchanged for $Q^*_F$. However, now the triangle inequality is violated:

$$Q^*_F(S_1, a, S_2) + Q^*_F(S_2, a, S_3) = -\alpha + \alpha \not\leq Q^*_F(S_1, a, S_3) \quad \square$$

This proves the criticality of Eq. 8, and that policy invariance of potential based shaping is insufficient to ensure the quasipseudometric property.

### 3.1.4 Projection

$Q^*_F$ has the same upper bound as $Q^*$, since $Q^*_F(s, a, g) = Q^*(s, a, g) - \phi(s, a, g) \leq 0$ by Eq. 8. Consequently, the MRN architecture needs no modification, specifically to Eq. 3, as the critic output is guaranteed to be non-positive despite potentially positive shaping rewards. However, $Q^*_F$ has a more informed lower bound:

$$Q^*_F(s, a, g) = Q^*(s, a, g) - \phi(s, a, g)$$
$$\geq -\frac{1}{1 - \gamma} - \phi(s, a, g)$$
$$= -\frac{\gamma^{d(s,a,g)/\eta}}{1 - \gamma} \qquad (13)$$

which we impose on the critic with $d_{ac}$ since it is bounded . Recent analyses (Gupta et al., 2022) have shown that projection informed by shaping effectively reduces the size of the state space for exploration, leading to improved regret bounds. We show the full algorithm for clarity in Alg. 1.

### 3.2 On-Policy Value Functions

In their critique of on-policy Q-function estimation methods for GCRL such as DDPG in continuous control tasks, Wang et al. (2023) show that *on-policy* Q-function may not be a quasipseudometric, even though the *optimal* Q-function is. However, their counterexample is an extreme policy that is unlikely to be encountered during on-policy iterations. In this section, we establish that on-policy Q-functions do indeed satisfy the triangle inequality (and hence meet the quasipseudometric criterion) if the policy makes a minimal progress toward the goal. We call such policies *progressive policies* and believe they are more relevant to on-policy Q-function estimation in GCRL. We first formalize the notion of progressive policies, specify our assumption, and finally show that the corresponding value functions satisfy the triangle inequality.

For notational convenience, we write $\mathbb{E}_{s' \sim T(.|s,a), a' \sim \pi(s')}$ simply as $\mathbb{E}_{s', a'}$. Note that the on-policy value function for a policy $\pi$ satisfies

$$Q^\pi(s, a, g) = R(s, a, g) + \gamma \mathbb{E}_{s', a'} \left[ Q^\pi(s', a', g) \right]. \qquad (14)$$

---

**Algorithm 1** GCRL with Dense Rewards

---
1: Randomly initialize MRN critic $Q_F$ and actor $\pi$
2: **for** $episode \leftarrow 1, 2, \ldots$ **do**
3:      Select goal $g \sim \rho_{\mathcal{G}}$, initial state $s \sim \rho_0$
4:      Select action $a \sim \pi(\cdot|s, g)$
5:      **for** $t \leftarrow 1, 2, \ldots$ **do**
6:          Execute action $a$ in state $s$ and receive reward $r$, next state $s'$
7:          Select next action $a' \sim \pi(\cdot|s', g)$
8:          Store tuple $(s, a, r, s', a', g)$ in replay memory $\mathcal{D}$
9:          $a, s \leftarrow a', s'$
10:      **end for**
11:      Sample minibatch $(s, a, r, s', a', g)$ from $\mathcal{D}$
12:      Relabel goals $g$ to $g'$ in the minibatch by HER with original reward $r$
13:      Use reward $(r + F(s, a, s', a', g'))$ in Eq. 2 to update MRN critic $Q_F$
14:      Clip $Q_F$ according to Eq. 13 if using arc-cosine distance $d_{ac}$
15:      Update actor $\pi$
16: **end for**

---

**Definition 1.** *The progress of a GCRL policy $\pi$ is given by*

$$\Delta^\pi(s, a, g) = \mathbb{E}_{s', a'}\left[Q^\pi(s', a', g)\right] - Q^\pi(s, a, g)$$

*for any $(s, a, g) \in \mathcal{S} \times \mathcal{A} \times \mathcal{G}$.*

We refer to $\Delta^\pi$ for the optimal policy as $\Delta^*$. We assume that the progress of $\pi$ is not unboundedly different from that of the optimal policy, i.e., the following holds for some $0 < \epsilon < \infty$

$$\epsilon \leq \Delta^*(s, a, g) - \Delta^\pi(s, a, g) \leq 2\epsilon. \tag{15}$$

Note that (i) $\epsilon$ does not need to be small, just finite; (ii) the counterexample in Wang et al. (2023) does not satisfy this assumption. Our main result of this section is:

**Proposition 3.** *Consider the goal-conditioned MDP $M_{GC} = (\mathcal{S}, \mathcal{A}, \mathcal{G}, T, R, \gamma, \rho_0, \rho_g)$. The on-policy value function $Q^\pi$ defined in Eq. 14 for any policy $\pi$ that satisfies Eq. 15 also satisfies the triangle inequality: $\forall x^1, x^2, x^3 \in \mathcal{X}$,*

$$Q^\pi(x^1, x^2) + Q^\pi(x^2, x^3) \leq Q^\pi(x^1, x^3).$$

**Proof:** From Eq. 14 we have,

$$\mathbb{E}_{s', a'}\left[Q^\pi(s', a', g)\right] = (Q^\pi(s, a, g) - R(s, a, g))/\gamma.$$

Then, using Eq. 15 and Def. 1, for either $z \equiv (x^1, x^2)$ or $z \equiv (x^2, x^3)$, the following holds:

$$\begin{aligned}
\Delta^*(z) - \Delta^\pi(z) &= \frac{Q^*(z) - R(z)}{\gamma} - Q^*(z) - \frac{Q^\pi(z) - R(z)}{\gamma} + Q^\pi(z) \\
&= (\frac{1}{\gamma} - 1)[Q^*(z) - Q^\pi(z)] \\
&\geq \epsilon \quad \text{(by Eq. 15).}
\end{aligned}$$

Adding for $z \equiv (x^1, x^2)$ and $z \equiv (x^2, x^3)$, we get

$$Q^\pi(x^1, x^2) + Q^\pi(x^2, x^3) \leq Q^*(x^1, x^2) + Q^*(x^2, x^3) - \frac{2\epsilon\gamma}{1 - \gamma} \tag{16}$$

But similarly for $z \equiv (x^1, x^3)$,

$$\Delta^*(z) - \Delta^\pi(z) = (\frac{1}{\gamma} - 1)[Q^*(z) - Q^\pi(z)] \le 2\epsilon$$

by Eq. 15. This gives $Q^*(x^1, x^3) \le Q^\pi(x^1, x^3) + \frac{2\epsilon\gamma}{1-\gamma}$. Finally, the result is obtained by combining this with Eq. 16 and noting that the triangle inequality holds for the optimal Q-value function, i.e., $Q^*(x^1, x^2) + Q^*(x^2, x^3) \le Q^*(x^1, x^3)$. $\qquad\square$

This result relies on the triangle inequality of the optimal value function as established before in (B. Liu et al., 2023) for sparse rewards and in Sec. 3.1 for dense rewards. But it does not have any dependence on whether $M$ is one-to-one or onto, hence the two cases $\mathcal{G} \equiv \mathcal{S} \times \mathcal{A}$ and $\mathcal{G} \not\equiv \mathcal{S} \times \mathcal{A}$ do not need to be distinguished. The result also does not assume any specific form of, or bounds on, the reward function. Hence it extends readily to shaped rewards as well, as long as the shaped value function respects the same upper bound (Sec. 3.1.4), $Q_F^\pi(.) \le 0$.

## 4  Related Work

Several value function representations have been proposed for GCRL over the last decade. Schaul et al. (2015) introduced the bilinear decomposition, later generalized to bilinear value networks (Yang et al., 2022) with better learning efficiency. Pitis et al. (2020) proposed the deep norm (DN) and wide norm (WN) families of neural representations that respect the triangle inequality. However, they are restricted to norm-induced functions, and are generally unable to represent all functions that respect the triangle inequality. By contrast, Possion Quasi-metric Embedding (PQE) (Wang & Isola, 2022) can universally approximate *any* quasipseudometric, thus improving upon DN/WN. However, as B. Liu et al. (2023) argue, PQE captures the restrictive form of first hitting-time when applied to GCRL, whereas MRNs capture the more general setting of repeated return to goal ($Q^*(g, g) \ne 0$), while preserving the UVFA property of PQEs. Durugkar et al. (2021) introduced a quasipseudometric that estimates the Wasserstein-1 distance between state visitation distributions, minimizing which is equivalent to policy optimization in GCRL tasks with deterministic transition dynamics. While they use the Wasserstein discriminator as a potential for reward shaping (as intrinsic motivation), our goal is different. We prove that dense rewards via shaping preserves the triangle inequality for the general class of potential based shaping, not just for the Wasserstein based quasipseudometric. Other recent architectures for GCRL use contrastive representation (Eysenbach et al., 2022) but without regard to quasipseudometric architecture, and quasipseudometric RL (QRL) (Wang et al., 2023) where temporal distances are learned, although it is unclear if it respects the triangle inequality in stochastic settings.

While the above literature on representation learning has been centered on expressive and flexible representations for GCRL, their analyses are generally restricted to sparse reward settings. In fact, past experimentation with dense rewards in GCRL have yielded negative results (Plappert et al., 2018). Plappert et al. (2018) argue that dense reward signals are hard to learn from because (i) arbitrary distance measures (e.g., Euclidean distance and quaternions for rotations) are highly non-linear; (ii) dense rewards bias the policy toward specific strategies that may be sub-optimal. Similar arguments also appear in (M. Liu et al., 2022). However, our setting overcomes these objections. First, we establish the sufficient condition (Eq. 8) for the triangle inequality that may not be satisfied by arbitrary distance measures, $\phi$, providing guidance on the contrary. And second, we use potential based reward shaping (Ng et al., 1999) which is policy invariant, hence strategically unbiased. However, we acknowledge the large body of work on reward shaping (Brys et al., 2014; Devlin & Kudenko, 2012; Knox & Stone, 2009; Tang et al., 2017; Van Seijen et al., 2017) of various types (e.g., count-based, intrinsic motivation, human advice, etc.) where careful, heuristic reward design is often employed to explicitly bias the policies.

## 5  Experimental Results

We use GCRL benchmark manipulation tasks with the Fetch robot and Shadow-hand domains (Plappert et al., 2018); see Fig. 2. MRN has been extensively compared with competitive baseline architectures and

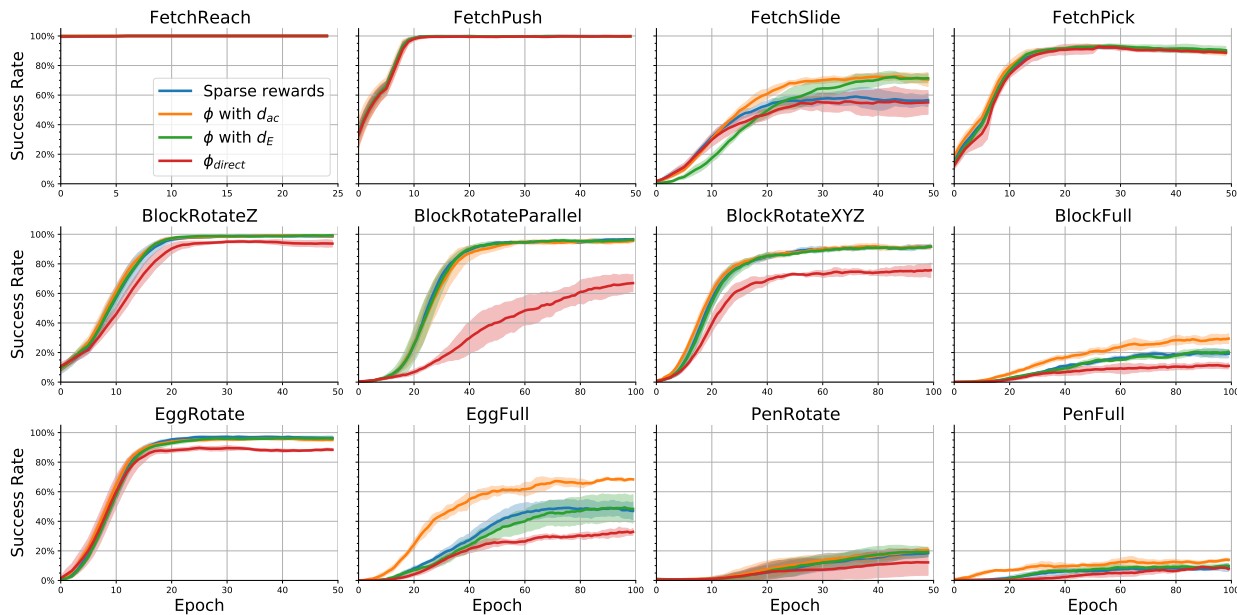

Figure 3: Comparison of MRN with sparse rewards vs. dense rewards with thee definitions of potential $\phi$. Learning curves are averaged over five independent trials, and one standard deviation bands are included. We see statistically significant improvement of performance due to dense rewards with $d_{ac}$ in 4 of the 12 environments, viz., FetchSlide, BlockFull, Eggfull and PenFull, and in 1 environment (FetchSlide) for $d_E$. There is no statistically significant deterioration in any environment for either $d_{ac}$ or $d_E$.

found to be superior, viz., BVN (Yang et al., 2022), DN/WN (Pitis et al., 2020), and PQE (Wang & Isola, 2022). Consequently, we focus on comparing against MRN with sparse rewards as the sole baseline. We experimentally evaluate the following hypotheses:

**Hypothesis 1:** Dense rewards may improve GCRL's sample complexity when using the MRN critic architecture. Specifically, the property of $Q^*$ function that MRNs capture—that it satisfies the triangle inequality—is preserved in the presence of shaped rewards with the new value function $Q_F^*$. Dense rewards may enable the less restrictive $Q_F^*$ to be learned more efficiently than $Q^*$.

**Hypothesis 2:** Plappert et al. (Plappert et al., 2018) found that dense rewards hurt RL performance in GCRL robot manipulation tasks. This negative result contradicts our Hypothesis 1. We conjecture that their application of dense rewards did not satisfy the required structure—specifically Eq. 8—which is why it failed. To confirm this contradiction, we verify that our dense reward setting does not deteriorate RL performance in any task using potential function based on either $d_{ac}$ or $d_E$. Additionally, to investigate the consequence of violating Eq. 8, we test a direct potential function $\phi_{\text{direct}}(s, a, g) = -\|M(s, a) - g\|$ that bypasses the distance function $d$. Note that as $\phi_{\text{direct}}$ is no longer comparable to $Q^*$, it may easily violate Obs. 1 and Eq. 8. Moreover, as it is potentially unbounded, we do not clip the corresponding critic (Eq. 13).

We use the MRN code repository publicly available at: `https://github.com/Cranial-XIX/metric-residual-network` which implements DDPG+HER with MRN critics for the GCRL manipulation tasks. We make two simple modifications: (1) add Eq. 6 to the default (sparse) reward function (Eq. 1) as shown on line 13 in Alg. 1; (2) use Eq. 13 to clip the critic's output as shown on line 14 in Alg. 1, but only for $d_{ac}$. No other changes were made to any algorithm or neural architecture. In particular, all parameter values (e.g. layer sizes) were unchanged, except the newly added parameter $\eta$ was set to 0.02 for $d_{ac}$ and 0.2 for $d_E$. These values were selected from the sets $\{0.01, 0.02, 0.03, 0.04, 0.05\}$ for $d_{ac}$ and $\{0.02, 0.2, 2.0\}$ for $d_E$, using performance improvement as the criterion. For each environment, 5 seeds were used for independent trials, as in (B. Liu et al., 2023). In each epoch, the agent is trained on 1000 episodes

| Environment | $\phi$ with $d_{ac}$ (%) | $\phi$ with $d_E$ (%) | $\phi_{\text{direct}}$ (%) |
|---|---|---|---|
| FetchReach | $59.098 \pm 1.444$ | $86.345 \pm 4.098$ | $8.523 \pm 0.203$ |
| FetchPush | $98.984 \pm 0.480$ | $46.860 \pm 1.071$ | $28.664 \pm 0.599$ |
| FetchSlide | $99.726 \pm 0.089$ | $95.851 \pm 0.248$ | $63.053 \pm 2.149$ |
| FetchPick | $97.907 \pm 0.819$ | $71.256 \pm 0.932$ | $26.057 \pm 0.889$ |
| BlockRotateZ | $96.453 \pm 1.052$ | $82.453 \pm 0.692$ | $4.693 \pm 0.182$ |
| BlockRotateParallel | $98.927 \pm 0.387$ | $81.996 \pm 0.681$ | $5.751 \pm 0.306$ |
| BlockRotateXYZ | $98.786 \pm 0.289$ | $91.542 \pm 0.371$ | $10.269 \pm 0.262$ |
| BlockFull | $99.705 \pm 0.115$ | $92.574 \pm 0.259$ | $40.727 \pm 2.390$ |
| EggRotate | $97.842 \pm 0.908$ | $85.228 \pm 1.104$ | $6.674 \pm 0.272$ |
| EggFull | $99.647 \pm 0.201$ | $95.662 \pm 0.560$ | $56.173 \pm 3.040$ |
| PenRotate | $97.955 \pm 1.066$ | $79.251 \pm 1.168$ | $20.911 \pm 0.964$ |
| PenFull | $97.994 \pm 0.760$ | $76.868 \pm 1.044$ | $21.576 \pm 0.586$ |

Table 1: How often the three potential functions satisfy Eq. 8.

and then evaluated over 100 independent rollouts with randomly sampled goals. The average success rates in these evaluations are collected over 5 seeds for baseline (sparse rewards) and the proposed method with three definitions of potential $\phi$, viz., with $d_{ac}$, with $d_E$ and $\phi_{\text{direct}}$. The results are plotted in Fig. 3. We also evaluate how well/often the three definitions of potential satisfy Eq. 8 w.r.t $Q^*$-values from the baseline. For this, we consider the $Q$-function returned from the last epoch of the baseline run as $Q^*$ [1], evaluate it over 1000 additional rollouts, and record the percentage of times that each $\phi$ is at least $Q^*$. These results are shown in Tab. 1. All experiments were run on NVIDIA Quadro RTX 6000 GPUs with 24 GiB of memory each and running on Ubuntu 22.04.

We see from Fig. 3 that indeed dense rewards with $d_{ac}$ and $d_E$ improve the sample complexity in some environments, to an extent that is statistically significant as shown with standard deviation bands. In particular, there is statistically significant improvement in 4 of the 12 environments, viz., FetchSlide, BlockFull, Eggfull and PenFull with $d_{ac}$, but only one environment (FetchSlide) with $d_E$. Not only is the sample complexity improved, but also higher quality policies are learned. This confirms Hypothesis 1. Furthermore, no statistically significant deterioration is observed in any environment with $d_{ac}$ or $d_E$, confirming Hypothesis 2. The sample complexity of $\phi$ with $d_E$ could conceivably improve with a more informed choice of $\eta$.

Fig. 3 shows that sample complexity is significantly worsened in 6 of 12 environments with $\phi_{\text{direct}}$, viz., BlockRotateZ, BlockRotateParallel, BlockRotateXYZ, BlockFull, EggRotate and EggFull. This is consistent with Tab. 1 which shows that $\phi_{\text{direct}}$ has a high propensity of violating Eq. 8 in most of these cases, which is further evidence of the criticality of Eq. 8.

## 6 Conclusion

We have presented generalizations of previous results on triangle inequality in the context of value functions in GCRL. Specifically, we have shown that the optimal value function satisfies the triangle inequality even when the reward function is densified with a particular class of shaping functions. Additionally, we have shown that the on-policy value functions also satisfy the triangle inequality if the underlying policy satisfies a certain progressive criterion. Both of these findings contradict previously published results in some ways, which emphasizes the importance of the nuanced conditions behind our results. Experiments in 12 benchmark GCRL tasks confirm that dense rewards either improve upon or match the sample efficiency of the sparse reward setting, but could worsen it if the key condition is violated. Future investigations could focus on more general classes of reward functions that preserve the quasipseudometric property of value functions and/or lend themselves to other, potentially more effective, architectures.

---

[1]This is an approximation and not strictly true.

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
