# OpenReview forum: "Quasipseudometric Value Functions with Dense Rewards"
_TMLR — Rejected by TMLR_

### Review · Reviewer_LnUe · 2025-02-16

**Summary Of Contributions:**

This paper demonstrates that the optimal value function adheres to the triangle inequality for a certain selection of the reward function.

Likewise, it establishes that the Q^pi function also satisfies the triangle inequality.

Through experiments, the authors show that dense rewards enhance sample complexity in 4 out of 12 tasks while maintaining performance in the remaining cases.

**Audience:**

Yes

**Claims And Evidence:**

No

**Requested Changes:**

It is overall unclear why the authors would like to investigate a very specific form of MDP and prove triangle inequality is satisfied. Although the most important criteria for this venue is that claims made in the submission supported by accurate, convincing and clear evidence, providing more context on the problem and motivations will attract more readers and make claims clearer and more solid.

The paper references the 2016 and 2017 works (DDPG and HER) when discussing the popular approach to solving GCRL. Could you include more recent references?

The definition of the variable L* is unclear, as it does not appear to be expressed as a function of input s.

Could you clarify the reasoning behind the chosen forms of F, phi, and d in the paper?

Can you explain the meaning of Q*_F and the notation  x_<t ?

The selection process for dense and sparse rewards in the experiments is unclear, as is the overall experimental setup (e.g., the definition of MDPs). Could you provide more details on the experimental design?

Additionally, it is unclear which algorithm is used in the experiments and how it is structured. The claim regarding improved sample complexity is unconvincing due to the lack of discussion on the algorithms, and it appears that only a single specific algorithm was tested.

Including a full page about the environments seems unnecessary, especially since the information is taken from another paper.

**Strengths And Weaknesses:**

Strength

Exploring the comparison between dense rewards and GCRL is an interesting direction.

The arguments presented in this paper are unclear to me.

Weakness

Both the proof and the experiment appear to lack rigor.

---

> ### Author Response · Authors · 2025-03-04
>
> Thank you for your feedback. We have revised the manuscript with changes shown in red color.
>
> We have added more context and motivations throughout the paper as advised.
>
> DDPG+HER, although from 2016,17, are still popular and commonly used in online GCRL as mentioned in (B. Liu et al., 2023). Also, much of the recent work has focused on offline GCRL. We have added a sentence with a new citation at the end of Sec. 2.2.
>
> Notation L* is now fully specified and $x_{<t}$ is explained. Other expressions, $F, \phi, d, Q^*_F$ are now explained in more details in Sec. 3.1.
>
> The MDPs/tasks studied in the experiments are specified in (Plappert et al., 2018) and we avoid repeating them to save space. Our parameter choice and experimental design are specified below Hypothesis 2 in Sec.5, where we have added further clarifications.
>
> We now show the algorithm as Algorithm 1. We have also included two additional potential functions: (1) $\phi$ with a different distance measure, viz., Euclidean distance $d_E$ (now introduced in Sec. 3.1); (2) direct potential $\phi_{\text{direct}}(s,a,g)=-||M(s,a)-g||$ that does not use a distance measure $d$ and can easily violate Eq. 8. These are now discussed at the end of Hypothesis 2. The results are now added to Fig. 3, with additional explanation in Sec. 5, and a new table (Tab. 1) is included to show how well the different potential functions satisfy Eq. 8.
>
> Fig. 2 is now smaller.

---

> > ### Comment · Reviewer_LnUe · 2025-03-15
> >
> > Thanks for addressing my questions!

---

### Review · Reviewer_THPX · 2025-02-25

**Summary Of Contributions:**

Previous works identified triangle inequality as a condition for optimal value function approximator under the setting of sparse reward in goal-conditioned reinforcement learning(GCRL) and correspondingly designed neural network architecture composed of a symmetric term and a residual term that better aligns with such quasimetric structure.

This work shows that the triangle inequality condition can be preserved in the dense reward setting with a simple choice of potential-based shaped rewards. Furthermore, the paper identifies the condition of making minimal progress towards the goal for on-policy value functions to satisfy the triangle inequality. They provide empirical evaluations to demonstrate that in the setting of dense rewards, GCRL achieves no worse performance than in the setting of sparse rewards.

**Audience:**

No

**Broader Impact Concerns:**

This work is mostly theoretical and no ethical implications of the work is of concern.

**Claims And Evidence:**

Yes

**Requested Changes:**

1. More explicitly demonstrate how promoting reward-shaping / dense reward setting in the context of goal-conditioned RL would be beneficial.
2. Discuss how the design/choice of the reward-shaping mechanism would affect the result, e.g., the triangle inequality condition, and if this limits the generality of the claims in the paper.

**Strengths And Weaknesses:**

**Strengths**
1. The paper generalizes previous work on value function approximators that satisfy the triangle inequality condition from the sparse reward setting to the dense reward setting.
2. The result complements previous work on multi-goal RL (Plappert et al., 2018), which found that goal-conditioned RL seems to work better in the sparse reward setting.

**Weaknesses**
1. The analysis is largely based on previous work, which makes the contribution seem limited.
2. Even though the reviewer mostly acknowledges the technical claims in this paper, the motivation of this paper remains a bit unclear. The authors claim their findings would open up opportunities to train efficient neural architectures with dense rewards, but the specific opportunities and how these opportunities would benefit GCRL are not obvious to the reviewer. Furthermore, it seems a bit contradictory to the motivation of the sparse reward setting, which is to avoid a complicated manually designed reward mechanism.
3. In the derivation, the paper applied a specific form of potential-based reward shaping so that the triangle inequality condition holds in the dense reward setting. It's unclear how sensitive the framework is to the way the reward is designed, as the authors have acknowledged in the paper that the result in (Plappert et. al. 2018) might have used a different reward design. This limits the generality of the claims made in this paper.

---

> ### Author Response · Authors · 2025-03-04
>
> Thank you for your feedback. We have revised the manuscript with changes shown in red color.
>
> Although some of our analyses indeed follow previous work closely, our analyses in Secs. 3.1.3, 3.1.4 and 3.2 are entirely new.
>
> We have now added clarification in Sec. 1 to motivate how promoting reward-shaping / dense reward setting could be beneficial specifically in the context of GCRL.
>
> To investigate how sensitive the framework is to the way the reward is designed, we have included two additional potential functions: (1) $\phi$ with a different distance measure, viz., Euclidean distance $d_E$ (now introduced in Sec. 3.1); (2) direct potential $\phi_{\text{direct}}(s,a,g)=-||M(s,a)-g||$ that does not use a distance measure $d$ and may easily violate Eq. 8. These are now discussed at the end of Hypothesis 2. The results are now added to Fig. 3, with additional explanation in Sec. 5, and a new table (Tab. 1) is included to show how well the different potential functions satisfy Eq. 8.  As $\phi_{\text{direct}}$ is a simple and popular reward shaping scheme, it is possible, even likely, that (Plappert et al. 2018) used this measure. Fig. 3 shows that $\phi_{\text{direct}}$ can indeed worsen sample complexity.

---

> > ### Comment · Reviewer_THPX · 2025-03-21
> >
> > Thank the authors for their reply. The quality of the manuscript has been improved

---

### Review · Reviewer_UctH · 2025-03-01

**Summary Of Contributions:**

This paper investigates the impact of dense reward shaping on GCRL, particularly focusing on the preservation of the triangle inequality — a key property for quasimetric value functions. Previous studies showed that the optimal value functions in GCRL have a quasimetric structure, but these studies largely focused on sparse rewards. By contrast, dense rewards were thought to deteriorate learning efficiency. The authors demonstrate theoretically that under specific conditions the quasimetric structure remains intact even with dense rewards. Empirical results across twelve benchmark robotics tasks indicate that dense rewards under these conditions can sometimes improve sample complexity without harming performance.

**Audience:**

Yes

**Claims And Evidence:**

No

**Requested Changes:**

1. Could the author provide an intuitive explanation of the metric residual network in Section 2.3? Why are the symmetric and asymmetric distances defined differently? And what does the term $\mu_{2i}$ in Equation 5 mean?

2. The quasimetric property and the triangle inequality of the value function are key concepts in this paper. However, the paper does not properly introduce them. I think they should be introduced in the background section.

3. In the experiments, I don't think the statement from Plappert et al. contradicts hypothesis 1. The fact that dense rewards hurt RL performance does not directly imply that dense rewards cannot be used with MRN for value function estimation.

**Strengths And Weaknesses:**

## Strengths

1. The author identifies a novel perspective to tackle the challenge of goal-conditioned reinforcement learning.

2. The author provides empirical evidence across multiple benchmark tasks, demonstrating that *appropriately* shaped dense rewards either enhance or maintain performance relative to sparse rewards.

## Weaknesses


**Presentation**
When reading this manuscript, I often felt something was missing. It would be better for the author to further improve the presentation. I list some key issues in this manuscript:

1. The terminology should be introduced before being used. For example, readers may feel unfamiliar with the term 'standard potential-based shaping rewards' in Eq. 6.

2. The motivation should be properly introduced. For instance, regarding Eq. 6, why is the function defined in this way? Why do you use the arc-cosine distance? You mentioned it is known to be a metric, but this explanation is insufficient.

3. The method is not systematically introduced. If possible, add a section called "Methods" and include pseudocode there.

**Experiments**
In Hypothesis 1, the authors state that dense rewards can be used in conjunction with the MRN architecture for estimating value functions. This argument cannot be supported only by the benchmark results. The authors may consider visualizing the values learned under different reward settings. More direct evidence should be presented.

In Hypothesis 2, the authors aim to demonstrate that the proposed dense reward setting does not deteriorate RL performance in any task. Empirically, the authors should propose at least 3–5 dense reward functions satisfying the stated condition to support this claim.

---

> ### Author Response · Authors · 2025-03-04
>
> Thank you for your feedback. We have revised the manuscript with changes shown in red color.
>
> We have now made sure to introduce terms before using them, e.g., policy in Sec 2.1, 'standard potential-based shaping rewards' around Eq. 6, potential, etc. We have also added more explanations in Sec. 3.1 about Eq. 6 and arc-cosine distance.
>
> We have now added pseudocode in Algorithm 1 with references in Secs. 3.1.4 and 5. As it is fairly self-explanatory, we have not added a "Methods" section to explain it.
>
> Hypothesis 1 was indeed malformed. We have now edited it.
>
> For Hypothesis 2, we have included two additional potential functions: (1) $\phi$ with a different distance measure, viz., Euclidean distance $d_E$ (introduced in Sec. 3.1); (2) direct potential $\phi_{\text{direct}}(s,a,g)=-||M(s,a)-g||$ that does not use a distance measure $d$. These are now discussed at the end of Hypothesis 2. The results are now added to Fig. 3, and a new table (Tab. 1) is included to further support Hypothesis 2.
>
> We have now added an intuitive explanation of the metric residual network in Section 2.3.
>
> We have now added a new Sec. 2.4 to explain the quasipseudometric property.
>
> The statement from Plappert et al. now contradicts the modified Hypothesis 1.

---

> ### Comment · Reviewer_UctH · 2025-03-12
>
> Thank you for your update. I have reviewed the manuscript and noticed improvements. The changes have enhanced the overall quality of the work.

---

### Decision · Action_Editor_TQ55 · 2025-04-01

**Recommendation:** Reject

**Comment:**

In their initial comments, the reviewers expressed several concerns about the original write-up, including:
- the quality of the presentation, the terminology used, and the paper’s organization,
- the motivation for studying the problem at hand,
- the rigor with which the technical aspects of the work were presented, as well as the clarity of their explanations,
- the rigor/depth of the experiments and the adequacy of the supporting evidence for the claims made,
- the scope of the work, which may limit the significance of its contributions.

While all reviewers acknowledge that the revised version represents a significant improvement---and some believe the changes made are sufficient---others maintain that the manuscript would still benefit from a major revision. These concerns extend beyond presentation, indicating the need for additional experimental results to more convincingly support the main claims. Below are some more concrete suggestions provided in the reviewers’ recommendations. I encourage the authors to incorporate these suggestions when revising their manuscript. Given these concerns, I recommend a major revision.

**Additional (more specific) suggestions provided by some of the reviewers:**

- The authors design a reward function $r_t + F(s, a, s', a', g')$ , where $F$ is defined in Equation 6 and requires a potential function $\phi$. In the experiments, the authors examine different potential functions ($\phi$ with $d_{ac}$, $\phi$ with $d_E$, and $\phi_{direct}$). However, $d_E$ is defined as a dense reward function in many environments and is not intended to serve as a potential function (see the Fetch environments in [1]). Therefore, the authors should compare their algorithm with the simple dense reward function directly, rather than treating it as a potential function. Ideally, the proposed reward function should outperform the simple dense reward.

- In Fetch environments such as FetchReach, FetchPush, FetchPick, and FetchSlide (Figure 3), it is reasonable to define the reward or potential function using the Euclidean distance, as the goal corresponds to the object's coordinates in 3D space. However, this approach is not appropriate in Block environments, where the goal includes both the position **and** orientation of the block. In such cases, the goal is represented by a 7-dimensional vector, and computing the Euclidean distance directly is not meaningful, not to mention the cosine distance (see the Robotics environments in [1]). While it is possible to train a policy for block manipulation using Euclidean distance as a reward or potential function, doing so raises theoretical concerns.

 - The environments tested in this manuscript typically have their own default dense reward functions. The authors should compare their proposed reward function against these default dense rewards. Furthermore, a possible extension could be to use a combined reward of the form $r_t + r_t^d$, where $r_t^d$ denotes the environment's default dense reward. Note that these default reward functions are not manually engineered; they are intuitive.

- In Figure 3 - BlockRotateParallel, the algorithm using sparse rewards appears to achieve higher performance. I suggest the authors use a thicker line to make it more distinguishable and avoid visual overlap.

 - It is not appropriate to use "triangle inequality" as the title of Section 3.

- The value function is not on-policy. In Algorithm 1, I agree that $(s, a, r, s', a', g)$ in Line 11 is on-policy data, however $(s, a, s', a', g')$ in Line 13 is not on-policy data since goal relabeling is used here. The current policy may not produce $a$ given $s$ and $g'$.

**Reference**: [1] Gymnasium-Robotics Documentation. https://robotics.farama.org/envs/fetch/pick_and_place/

**Audience:**

This paper would be of interest to researchers working on goal-condition RL. It may also be of interest to the RL community in general, and more specifically researchers interested in reward design and reward shaping.

**Claims And Evidence:**

The paper studies the utility of dense rewards in goal-conditioned RL (GCRL). More specifically, the paper provides conditions under which the optimal value function, as well as on-policy value functions, are quasipseudometrics (i.e., they satisfy the triangle inequality) in dense reward settings. It also experimentally compares the sample complexity of variants of the MRN approach---which relies on this property---under both dense and sparse rewards. As outlined in my comments below, the reviewers indicated several issues with the original submission (e.g., the clarity/organization of the paper, the motivation of the work, the rigor/depth of the experiments), some of which have been alleviated in the revised version of the manuscript. Further details are provided below.

**Resubmission Of Major Revision:**

The authors may consider submitting a major revision at a later time.